**SOFTWARE**

# Sequre: a high-performance framework for secure multiparty computation enables biomedical data sharing

Haris Smajlović[1], Ariya Shajii[2], Bonnie Berger[2*], Hyunghoon Cho[3*] and Ibrahim Numanagić[1*]

*Correspondence:
bab@mit.edu;
hhcho@broadinstitute.org;
inumanag@uvic.ca

[1] Department of Computer Science, University of Victoria, Victoria, BC, Canada
[2] CSAIL, MIT, Cambridge, MA, USA
[3] Broad Institute of MIT and Harvard, Cambridge, MA, USA

## Abstract

Secure multiparty computation (MPC) is a cryptographic tool that allows computation on top of sensitive biomedical data without revealing private information to the involved entities. Here, we introduce Sequre, an easy-to-use, high-performance framework for developing performant MPC applications. Sequre offers a set of automatic compile-time optimizations that significantly improve the performance of MPC applications and incorporates the syntax of Python programming language to facilitate rapid application development. We demonstrate its usability and performance on various bioinformatics tasks showing up to 3–4 times increased speed over the existing pipelines with 7-fold reductions in codebase sizes.

**Keywords:** Genomic privacy, Secure multiparty computation, Domain-specific language

## Background

Privacy concerns present a key hurdle in genomic data-sharing efforts. Genomic data leaks are not only irreversible because one's genetic sequence cannot be changed, but their potential harm also extends to the genetic relatives of the individuals whose data is leaked. Traditional approaches to privacy protection such as de-identification and access control, as described in policies such as the Health Insurance Portability and Accountability Act (HIPAA) in the USA and the Personal Information Protection and Electronic Documents Act (PIPEDA) in Canada, provide limited guidance for responsible sharing of genomic data due to inability to fully de-identify such data. Furthermore, novel privacy attack surfaces continue to be discovered, exacerbating these concerns [1–4]. Keeping the data in silos and imposing strict data access and sharing restrictions either prevent or slow down biomedical research, which increasingly depends upon access to large datasets [5].

Recent advances in secure computation technologies offer a promising approach for mitigating the privacy concerns associated with data sharing [6]. These technologies generally enable computation on private data—in an encrypted form—without disclosing the sensitive information to anyone involved. A prominent such approach is secure multiparty computation (MPC) [7], which distributes the private data to multiple computing parties in a form that does not reveal any sensitive information to either party, but allows all parties to interactively carry out the desired computation without revealing the underlying data. As the private data is kept confidential throughout the analysis, this approach allows private data held by multiple parties to be securely and jointly leveraged without disclosing the raw data. Recent studies have demonstrated the practical applicability of MPC for a range of computational genomics and biomedical research workflows [8–13]. Note that MPC in principle allows arbitrary computation over private data with formal security guarantees, and its lower computational cost compared to other frameworks (such as homomorphic encryption) makes it an appealing solution for analyzing large-scale datasets, which are common in biomedical domains [7, 8, 11, 13–16]. Unlike trusted execution environment (TEE) technologies for secure computation [17], MPC does not require specialized hardware for its application.

However, the practical application of MPC has been stymied by the high cost of developing efficient MPC protocols with minimal computational overhead. The distributed nature of MPC implies that each data operation, such as multiplication of two secret numbers, needs to be performed in a coordinated manner across different parties, increasing the complexity of the computation compared to its non-secure counterpart. This overhead can make even simple algorithms many orders of magnitude (e.g., $100\times$ or more) slower than their non-secure counterparts [18]. Furthermore, existing non-secure pipelines cannot be easily ported to secure environments since MPC frameworks typically require (i) a near-complete reimplementation of existing algorithms using only low-level MPC routines and (ii) a manual optimization of the algorithms to improve MPC performance. Such optimization can obfuscate the original intent of the code and sacrifice readability and maintainability, thus making the subsequent development and code reviews (needed for security assurance and compliance) tedious and prone to oversight. These limitations are in part due to the fact that the existing MPC frameworks (e.g., [19]) are implemented as custom libraries of low-level MPC operations that cannot be efficiently composed and optimized [20], or as domain-specific languages with ad hoc syntax and limited expressiveness that increase the difficulty of MPC pipeline development and maintenance [21].

Here we introduce Sequre, a Python-like, high-performance domain-specific language for developing secure MPC algorithms. Sequre uses standard Python syntax and semantics to ease the development of secure pipelines and the transformation of existing code into MPC equivalents. On the other hand, Sequre builds upon Seq [20], a recently introduced framework for compiling Pythonic codebases, and a set of newly developed compile-time code analysis strategies and optimizations to improve performance even beyond that of the fastest MPC frameworks in C/C++. Sequre's novel optimizations utilize Seq's *intermediate representation* (IR) of the Python source code (i.e., a logical representation of a program's execution flow that can be statically analyzed) to remove unnecessary computation and to select the best MPC routines and optimization

approaches for each computational step. As a result, Sequre enables high-performance and simple codebases that do not require extensive MPC-related modifications.

We demonstrate Sequre's performance and usability by employing it to implement various bioinformatics pipelines, including genome-wide association study (GWAS) [9], drug-target interaction (DTI) inference [10], and metagenomic binning [22, 23]. To our knowledge, a secure MPC protocol for metagenomic binning has not been previously developed. We implemented each pipeline in only 80–160 lines of high-level Python code whose functionality is equivalent to the original algorithm. Compared to the existing state-of-the-art pipelines, we achieved up to 7× reduction in code length. Furthermore, the overall execution time of these pipelines was reduced 3–4×, and the network utilization was also 17% lower. Where possible, we also compared Sequre to an existing Python-based MPC framework, PySyft (SyMPC) [19], and showed that various machine learning tasks can be performed 2× faster while providing comparable security guarantees. We also micro-benchmarked Sequre's performance and compared it to ten existing MPC frameworks using a standard benchmark suite [21]. Sequre achieved the best runtime performance in the majority of cases while being one of the easiest frameworks to use.

We expect Sequre to enable practitioners without expertise in MPC and cryptography to easily write efficient MPC algorithms for various biomedical workflows. Furthermore, the improved readability and usability of Sequre programs can simplify sharing and maintenance of these tools. Thus, Sequre could facilitate the use of secure computation technologies and, as a result, broaden data sharing and collaboration efforts in biomedicine.

## Results

### Overview of Sequre

Sequre is a high-performance framework for the development and deployment of secure multiparty computation (MPC) pipelines (Fig. 1; see Additional file 1: Section 2 for an overview of MPC). It takes in a computational pipeline written in the syntax of the widely used Python language and compiles it to an equivalent MPC program. The source code of the pipeline is statically analyzed to detect code blocks that need to be replaced with secure computation routines, as well as to discover and apply MPC-specific compile-time transformations and optimizations that can speed up the execution. The final output includes optimized executable programs that can be deployed by a group of computing parties to securely perform the desired computation on private data. The high-level source code written in Sequre can be easily reviewed by the involved entities to understand and agree upon the workflow before the execution.

Sequre's compiler pipeline and automated optimization techniques are illustrated in Fig. 2. Most importantly, arithmetic expressions are analyzed and restructured at compile-time to minimize the computation and network overhead. For example, the compiler will expand a series of expressions into a polynomial and invoke the optimized MPC routine for polynomial evaluation to minimize the network overhead. If the expanded polynomial is highly complex, the compiler will perform a static code analysis to find hotspots where auxiliary MPC computation can be cached—a procedure that would otherwise require significant manual intervention. Additionally, Sequre looks for

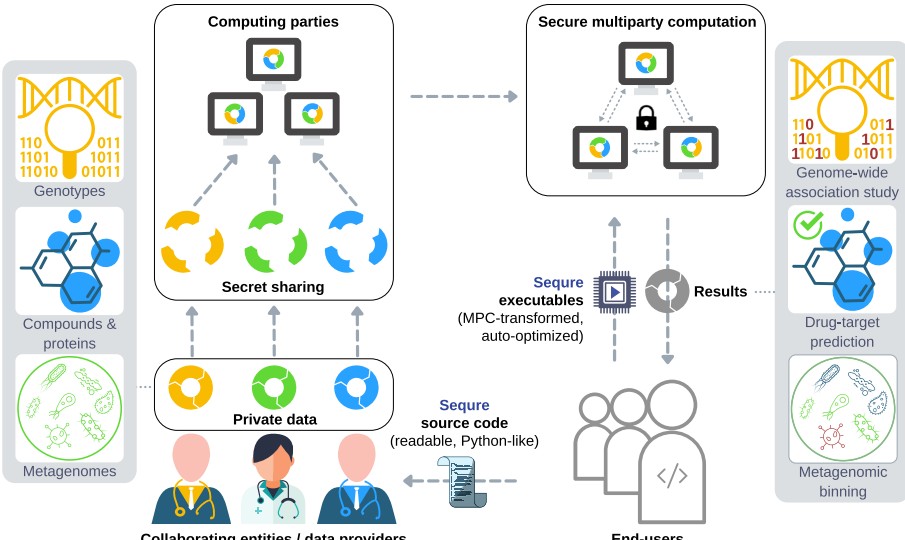

**Fig. 1** Overview of Sequre. Secure multiparty computation (MPC) enables collaborative analysis of sensitive private data—such as patient genomes and proprietary pharmaceutical datasets—without disclosing the data to anyone other than the respective *data providers*. This functionality has the potential to allow various stakeholders in biomedicine (e.g., academic, clinical, or commercial labs) to more broadly share sensitive data for a range of biomedical applications (e.g., genetic association studies, drug-target prediction, and metagenomic profiling). MPC enables secure collaboration by dividing sensitive data into encrypted shares and distributing them across multiple *computing parties* (may be the data providers themselves), leveraging a cryptographic technique known as *secret sharing*. Securely performing computation over the distributed encrypted data necessitates complex and specialized computational protocols, which often incur a significant performance overhead. Sequre addresses this challenge to accelerate MPC development and deployment by automatically converting programs written in a readable, high-level Python-like language into high-performance MPC programs, thus enabling both faster development and review cycles, as well as faster execution. We envision a workflow where the *end-users* (e.g., collaborating researchers) use Sequre to rapidly develop and agree upon on a pipeline for a secure collaborative study, then deploy the optimized executable programs produced by Sequre to computing parties for execution. Finally, the results of the collaborative analysis are returned to the end-users, revealing insights from the combined data that individual entities could not obtain otherwise

specific algebraic patterns (e.g., secure matrix and fixed-point arithmetics) and substitutes them with MPC-efficient alternatives.

In the following, we demonstrate three applications of Sequre in different domains, including medical genetics (GWAS), pharmacogenomics (drug-target interaction prediction), and metagenomics (taxonomic binning). The first two are reimplementations of recently published MPC solutions [9, 10], while the last (metagenomic binning [22, 23]) illustrates a novel MPC implementation of a common biomedical task from scratch. For comparison with existing tools, we evaluate all methods using a single thread and based on the same level of security (see Additional file 1: Section 5 for details). Our results illustrate Sequre's practical utility, performance, and usability.

### Secure genome-wide association studies

One of the first practical demonstrations of secure computation in genomics was for genome-wide association studies (GWAS) [8, 9, 11, 24]. GWAS aims to identify genetic variants that are statistically correlated with phenotypes of interest (e.g., biological traits or disease status). For example, Cho et al. [9] introduced an MPC

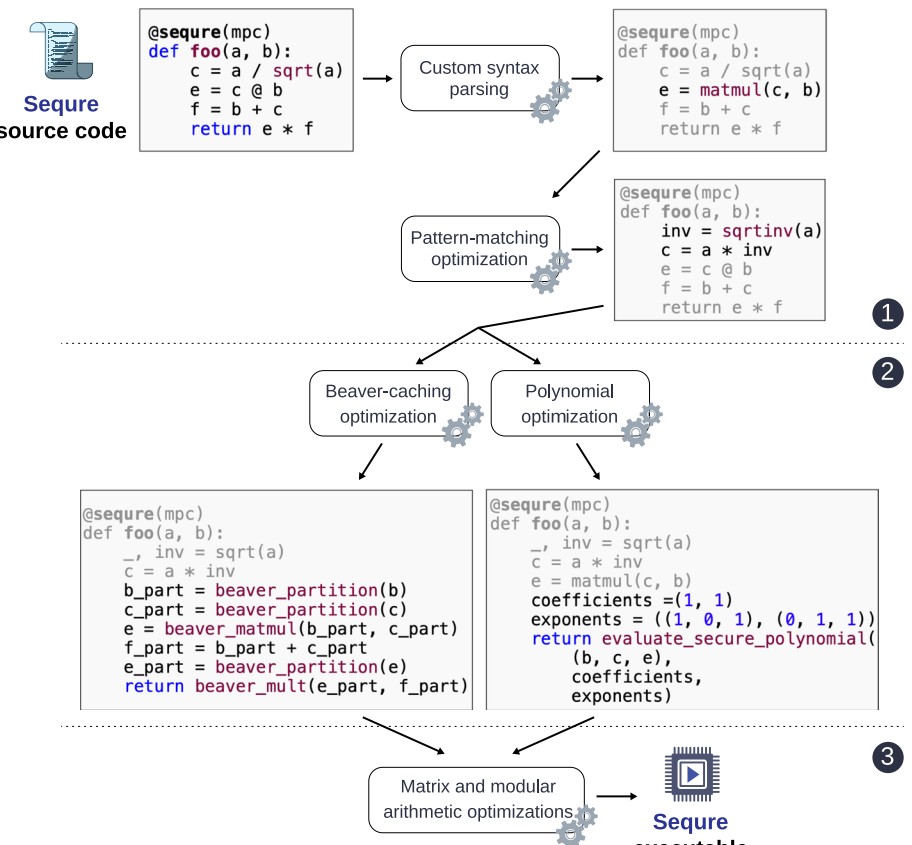

**Fig. 2** Sequre's automatic compiler optimization workflow. Sequre transforms an analysis pipeline written in standard Python language into equivalent secure MPC programs through a set of compiler analysis and optimization modules. From the end-user's perspective, the high-level syntax of Sequre is agnostic to MPC, because the individual operations are automatically replaced with the corresponding MPC routines by Sequre during compilation. The transformed code is then analyzed by the compiler in several passes to optimize the performance of the MPC pipeline. More specifically, Sequre's optimizations include (1) replacing common code patterns with more efficient equivalents under the MPC setting, (2) restructuring arithmetic expressions to minimize redundant computation and the network overhead incurred by the *Beaver partitioning* operations, and (3) applying a set of higher-level performance optimizations, such as faster modulus and matrix operations. The resulting program is a high-performance executable ready to be executed by computing parties to carry out the desired analysis. The optimization techniques of Sequre are described in the "Methods" section and Additional file 1: Section 4

solution written in more than 1000 lines of carefully optimized C/C++ code, encompassing all the standard steps of a GWAS: quality control filtering (to control missing rates, allele frequencies, and Hardy-Weinberg equilibrium), population stratification analysis through principal component analysis (PCA), and linear regression-based association tests. Despite extensive optimizations, this pipeline was estimated to require 80 days to securely perform GWAS on a million individuals and half a million single-nucleotide polymorphisms (SNPs), illustrating the overhead of MPC for complex operations such as GWAS.

We developed Sequre-GWAS, a reimplementation of the aforementioned MPC-based GWAS pipeline [9] in Sequre. Our implementation consisted of only 160 lines of high-level Python code, representing over 7× reduction in code length. We

observed a 3.7× decrease in the overall runtime of GWAS with a comparable network utilization on the lung cancer dataset from [9]. Further, we observed consistent speedup factors for varying dataset dimensions (Additional file 1: Section 3), based on which we estimate a runtime of 3 weeks for a million-individual study using Sequre-GWAS, in contrast to nearly 3 months reported in the original publication. The analysis results retain the same accuracy even after the transformations and optimizations automatically applied to the pipeline by the Sequre compiler (Additional file 1: Section 3).

The breakdown of the performance improvement achieved by Sequre is shown in Fig. 3. MPC-related optimizations, such as caching the intermediate results of secure multiplications and adjusting the precedence of operators to be executed in an MPC-friendly manner—both manually optimized in prior work—are automatically performed by our compiler. Not only Sequre reproduced all such optimizations in the original code, but also it found 4% more hotspots in the original codebase that could be further optimized. These optimizations alone, together with modular arithmetic optimizations, improved the overall GWAS runtime by 1.64×. Finally, automatic conversion from the

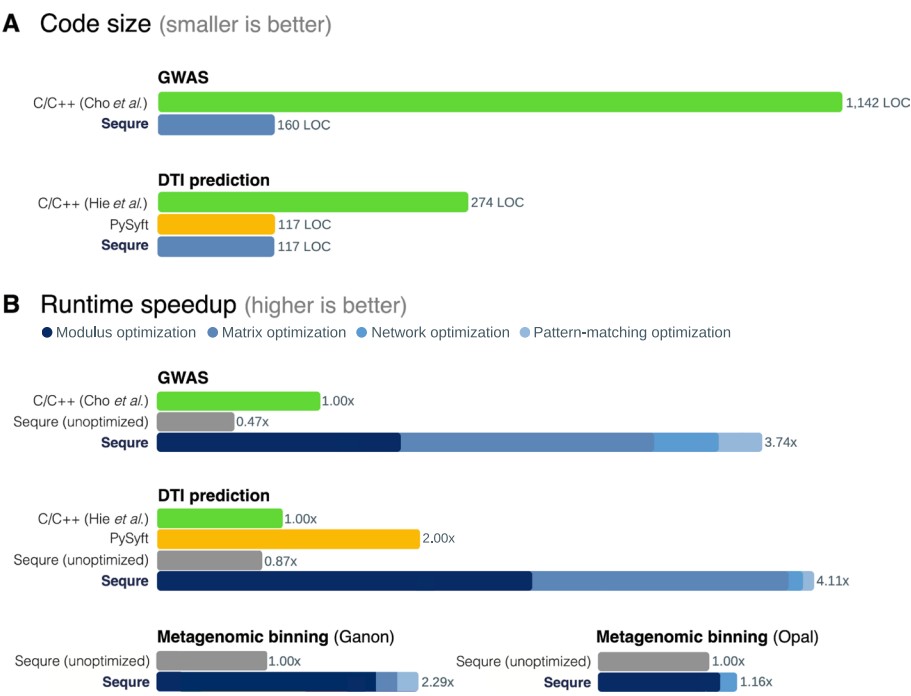

**Fig. 3** Sequre's usability and performance improvements in three biomedical applications. Sequre's automated code transformation and optimizations reduce **A** code complexity (number of the lines of code [LOC] in the implementation) and **B** runtime (execution time in seconds). We show Sequre's improvements in these metrics in three applications: genome-wide association studies (GWAS), drug-target interaction (DTI) prediction, and metagenomic binning. For metagenomic binning, we consider two recent algorithms, Ganon [22] and Opal [23]. We implemented the analysis pipeline for each application in Sequre and compared with the version without the compiler optimizations as well as implementations in existing frameworks where applicable (C/C++ and PySyft). Note that the C/C++ baselines refer to the recently published, manually optimized MPC implementations of GWAS [9] and DTI prediction [10]. There is no prior MPC implementation for metagenomic binning. Contributions of individual optimization modules in Sequre are shown in different colors within a bar. Sequre generates high-performance MPC programs while allowing them to be easily and compactly written in standard Python language

finite (Galois) field-based to the ring-based MPC protocols (Additional file 1: Section 4) resulted in an additional 2.3× speed-up.

### Secure drug-target interaction prediction

Another promising application of secure computation is drug-target interaction (DTI) prediction. The goal of DTI prediction is to uncover novel interactions between drug molecules and putative protein targets. The inference method takes a set of compounds and a set of targets as input and outputs the probability of interaction between new compound-target pairs. While the feature representations vary across the existing methods, the inference is increasingly performed using neural networks [10, 25–27].

Unfortunately, many existing pharmacological datasets are held by labs and commercial companies that are unable to share data due to intellectual property concerns. This problem was recently addressed by a privacy-preserving DTI prediction pipeline that protects the individual training datasets from the involved entities [10]. The pipeline begins with the data preprocessing step, where the chemical compounds and proteins are encoded as feature vectors by the data holders and securely shared with the untrusted computing parties. The second step of the pipeline—secure model training—uses these features to securely train a neural network using a secure MPC protocol (Additional file 1: Section 3). The existing method for model training takes up to 4 days for one million drug-target pairs [10]. After the training, DTI inference continues in a similar fashion: the data owners encode their drug-target pairs locally and securely evaluate the pre-trained neural network on the new inputs to obtain the predictions.

To demonstrate that machine learning tasks, such as neural network training, can be efficiently developed using our framework, we reimplemented the DTI prediction pipeline in Sequre (Sequre-DTI). We observed improvements similar to the GWAS task: 2–3× reduction in code length (from 274 to 117 lines of code), more than 4× faster execution times, and more than 34% reduction in network usage (Fig. 3; Additional file 1: Section 3). For example, 4 days of training time could be reduced to less than a day. The impact of individual optimizations in Sequre-DTI followed the same patterns as in the GWAS example.

We also implemented the DTI inference procedure in SyMPC (PySyft) [19], an existing Python framework specialized in deep learning MPC pipelines. While Sequre-DTI and the PySyft implementation were comparable in both accuracy and code length (Additional file 1: Section 3), Sequre-DTI was more than 2× faster (Fig. 3). This improvement is despite the fact that SyMPC employs function secret sharing (more restrictive but theoretically faster than Sequre's approach) and 64-bit data types that incur smaller CPU and network overhead than Sequre's 128-bit data types, thus highlighting the effectiveness of Sequre's automated optimizations.

### Secure metagenomic binning

To demonstrate that new analysis pipelines can be easily developed from scratch with Sequre, we turn to metagenomic binning, a central task in the analysis of microbiomes for which practical MPC algorithms have not yet been developed. The goal of this task is to identify and quantify the organisms present in a sequenced metagenomic sample by classifying reads and assigning them to reference genomes, resulting in a "metagenomic

profile" of the sample that can be used for various health-related tasks [28–31]. Despite the need for a service where the user can upload their samples and receive the binning results with respect to the existing reference datasets (which may be proprietary), privacy risks have limited the utility of such services. For example, a sequenced metagenomic sample may include the host's genetic sequence, as well as viral sequences potentially indicating the infection status of the host [32]. In some cases, even the metagenomic profile alone can identify the individual and disclose information about the host's behavior and environment [33].

There exists a wide variety of metagenomic binning methods and pipelines developed in the non-secure context [34]. Here, we used Sequre to implement two state-of-the-art metagenomics binning pipelines: Ganon [22], based on Bloom filters, and Opal [23], based on locally sensitive hashing and machine learning classifiers. We refer to our MPC implementations of Ganon and Opal as Sequre-Ganon and Sequre-Opal, respectively.

Given a sample including reads to be classified, Sequre-Ganon extracts sequence features from the reads and secretly shares them with the computing parties (Additional file 1: Section 3). The features are then securely queried against the index in the form of an interleaved Bloom filter [35]. The query results—the probability of a read belonging to a given bin (or reference)—are used to determine the most likely classification for each read. This procedure is implemented in a secure manner using oblivious array data structures [36], efficiently supported by Sequre. On the other hand, Sequre-Opal uses machine learning classifiers (logistic regression or support vector machines) for the same task. It also begins by encoding the reads as feature vectors, this time through Gallager coding [37], and secretly sharing the data with computing parties. We implemented MPC versions of both training and inference steps that use binary classifiers with hinge loss and stochastic gradient descent optimization.

Despite the complexity of both algorithms, Sequre-Ganon and Sequre-Opal are implemented as compact high-level Python programs in 113 and 80 lines of code, respectively. The code did not include any manual MPC-related optimization, thus being effectively identical to the non-secure counterparts. We evaluated our methods on the Opal benchmark dataset [23], which includes 10 reference bacterial genomes sequenced at 15× coverage as well as a classification test data including 10,000 reads of length 65 bp. Sequre-Ganon took 18.5 h to perform the classification—the task that otherwise takes less than 10 s in offline, non-secure setting, while Sequre-Opal took 3 h for both training and classification. As a reference, the non-secure Opal run terminates in less than 10 min (Additional file 1: Section 3). The large runtime difference is due to Sequre-Ganon's use of Bloom filters, which incurs a considerable overhead in the MPC setting: privately answering a query takes $O(n)$ time, where $n$ is the size of the data structure, compared to $O(1)$ time needed in the non-secure setting for the same task. The algorithm of Sequre-Opal is less affected by the algorithmic complexity changes induced by MPC protocols. Nonetheless, we note that Sequre still optimized the naïve MPC implementations of both methods. The compiler optimizations made Sequre-Ganon 2.29× faster, and Sequre-Opal 1.16× faster (Additional file 1: Section 3). Finally, we note that Sequre-Ganon and Ganon had identical accuracy as expected. This was also the case for Sequre-Opal and Opal based on a comparable choice of models and parameters (Additional file 1: Section 3).

### Sequre and other MPC frameworks

Many other MPC frameworks have been introduced in recent years [18, 19, 38–46] (see [21] for a survey of existing MPC frameworks). Each framework offers some novelty concerning security, expressiveness, or performance; however, most of the available frameworks are not yet ready for practical use due to performance limitations [21]. Sequre emphasizes practicality and optimizes expressiveness and performance by adopting an honest-but-curious security model based on the additive secret sharing scheme [7] and leveraging various optimizations based on compile-time static code analysis for improved performance. It operates under what we view as middle-ground security constraints, providing a rigorous notion of security based on the properties of secret sharing, while introducing additional requirements to enable efficient performance, namely an auxiliary party—known as a *trusted dealer*—that generates correlated randomness to be used in the main protocol to greatly accelerate the computation. This model is also known as the server-aided model of MPC, and it requires that the collaborating entities identify an independent, trustworthy actor to assume this role. However, many bioinformatics applications with large-scale datasets currently necessitate this modification to achieve practical performance. Nevertheless, disabling a trusted dealer at the expense of performance can be done, should one wish to do so. Additionally, as discussed in Cho et al. [9], the existing MPC scheme can be strengthened to allow both malicious and semi-honest adversaries, which relaxes the security assumptions by allowing the parties to deviate from the protocol at the expense of having worse performance.

Out of the available MPC frameworks, we selected and benchmarked ten mature and actively maintained MPC frameworks that are comparable to Sequre. We evaluated usability (as measured by code length), overall runtime and network utilization. It should be noted that not all frameworks operate under the same computational model and security constraints. For example, four evaluated frameworks use garbled circuits which support only two parties and the evaluation of Boolean circuits [7], while the other six operate under the same or, in some cases, slightly weaker models (e.g., honest-majority setting) compared to Sequre. For comparison, we used the closest MPC paradigm and parameter settings across the tools to the extent possible (Additional file 1: Section 3).

In terms of usability, we found Sequre to be 3× more expressive on average, measured by the number of lines of code required to implement the benchmark (fewer means more expressive) (Additional file 1: Section 3). Sequre is also one of the few frameworks that do not require learning a new language or framework: a single-line decorator is sufficient for Sequre to convert a normal Python code into its secure equivalent. Performance-wise, Sequre was on average 100× faster than the other frameworks (despite excluding the four outlier test cases where Sequre was from 1600× to 32,941× faster than its counterparts). More precisely, Sequre was up to 250× faster in 9 test instances based on similar security models, and up to 1117× faster in 8 test instances involving comparison with a different MPC paradigm (garbled circuits). Even when compared with frameworks with more limited security guarantees aimed at faster performance, Sequre was 3–9× faster over 9 test instances. The only cases where Sequre was slower (2.5–5.5×) were the four comparisons that evaluated the performance of oblivious data structures (e.g., secure array access), for which more efficient, specialized routines have been implemented in existing frameworks. Note that, given the modularity of Sequre,

more efficient subroutines such as these can be continuously integrated to further improve performance while maintaining the expressiveness of the Sequre language. The performance comparison between Sequre and three other frameworks that are similar to Sequre in terms of design and features is presented in Table 1. An extended cross-comparison between Sequre and ten other frameworks is provided in Additional file 1. We also note that because Sequre is based on Seq [20], it also provides a wide range of domain-specific features and routines for efficiently processing genomic datasets (e.g., sequence operations), which can be seamlessly integrated with the MPC portions of the analysis pipeline.

## Discussion

We note that there are inherent limitations to what can be achieved by the automated compiler optimization of Sequre. For example, the performance difference between the two Sequre tools for metagenomic binning (Sequre-Ganon and Sequre-Opal) illustrates how the performance of a program pivotally depends on the underlying algorithmic choices. To an extent, the user still needs to remain engaged in exploring different implementation strategies in order to obtain the most efficient tool for the desired task. The fact that Sequre allows the user to program in Python without any special consideration for MPC greatly simplifies and accelerates this development process. Providing a library of high-level routines that are commonly used in biomedical analyses (e.g., basic statistical models) may further reduce the user's burden on the algorithmic side and thus is a meaningful direction for future work.

Sequre can easily be extended to incorporate novel MPC protocols, frameworks, and optimization techniques. Sequre's approach can also be used to target other privacy-enhancing technologies such as homomorphic encryption or hardware-based trusted execution environment (TEE) technologies, which present unique challenges. Finally, practitioners in other fields beyond biomedicine can use Sequre to develop secure data

**Table 1** A cross-comparison between Sequre and three state-of-the-art MPC frameworks. Frameworks were benchmarked for expressiveness (in terms of lines of code (LOC)) and runtime over multiple MPC setups. The best runtimes per benchmark per setup are bolded. Some variants are not supported (marked with ⊥). A complete listing of cross-comparison against another seven frameworks is provided in Additional file 1

|  | Framework | LOC | Runtime (ms) | | | |
|---|---|---|---|---|---|---|
|  |  |  | **128bit$\mathbb{Z}_p$** | **128bit$\mathbb{Z}_{2^k}$** | **64bit$\mathbb{Z}_p$** | **64bit$\mathbb{Z}_{2^k}$** |
| `mult3` | MP-SPDZ | 4 | 1.0 | 0.9 | 0.7 | 0.6 |
|  | MPyC | 8 | ⊥ | ⊥ | 0.9 | ⊥ |
|  | Sharemind | 4 | ⊥ | ⊥ | ⊥ | 2.8 |
|  | Sequre | 4 | **0.2** | **0.1** | – | – |
| `innerprod` | MP-SPDZ | 7 | 78 | 45 | 77 | 44 |
|  | MPyC | 7 | ⊥ | ⊥ | 4,200 | ⊥ |
|  | Sharemind | 4 | ⊥ | ⊥ | ⊥ | 20 |
|  | Sequre | 4 | **24** | **17** | – | – |
| `xtabs` | MP-SPDZ | 24 | 70 | **20** | 40 | 15 |
|  | MPyC | 9 | ⊥ | ⊥ | 700 | ⊥ |
|  | Sharemind | 15 | ⊥ | ⊥ | ⊥ | 2500 |
|  | Sequre | 9 | **50** | 95 | – | – |

analysis pipelines. Our work provides a key tool for broadening data sharing and collaboration in biomedicine.

## Conclusions

We presented Sequre, a performant and user-friendly tool for developing privacy-preserving software for biomedical data analysis. Sequre introduces a compiler that transforms a high-level Python script to a secure MPC program while applying a variety of sophisticated code optimizations without manual intervention. This allows practitioners without the expertise in MPC to develop and use efficient MPC software. Our results on diverse applications demonstrate the usability of Sequre as well as its state-of-the-art performance, often outperforming carefully optimized published tools from prior works.

## Methods

### Sequre at a glance

Sequre is a high-performance secure multi-party computing (MPC) compiler framework consisting of a MPC library and a set of domain-specific compile-time transformation and optimization passes that can detect various MPC operations in the source code and automatically simplify and optimize them. Sequre uses Seq [47], a compiler framework for building statically typed high-performance languages that use Python's syntax and semantics. Thanks to Seq, Sequre is able to combine the ease of Python with the performance of C/C++.

Sequre follows the fundamental principle that the code can be optimized by the compiler automatically through a set of custom analysis and compile-time optimizations that utilize domain-specific knowledge. This principle has been successfully applied to languages and optimization toolkits in various domains including GPU computing [48], image processing [49], deep learning [50], tensor computing [51], parallel computing [52], and recently, bioinformatics [20, 53].

A Sequre pipeline is written in a dialect of Python that can be statically type checked [20]. Secure MPC procedures—i.e., blocks of code meant to operate on securely shared data in a distributed fashion—are annotated via the "`@sequre`" decorator. Code annotated with this decorator is automatically converted to a secure MPC routine by transforming each operation to the MPC equivalent implemented in Sequre's standard library. This library supports common arithmetic, Boolean and linear algebra operations, and shares the same semantics as Python's standard library and the NumPy library [54]. The transformed source code is then statically type checked and transformed to a Seq intermediate representation (IR), a starting point for all further analysis and optimization passes. After applying a basic set of general-purpose Python code optimizations [47], Sequre performs additional MPC-related optimizations, which aim to reduce network utilization and runtime performance of the pipeline. Optimized IR is then translated to LLVM IR and subsequently handled by the LLVM framework [55] that applies additional set of both general-purpose and MPC-related performance optimizations and facilitates the final machine code generation.

The final result is a highly optimized executable that can be deployed by a set of computing parties to perform the desired computation on the private data, as well as the original high-level source code that can be easily understood by involved entities (see Additional file 1: Section 1 for an example of a pipeline written in Sequre).

**Sequre's MPC framework**

Sequre uses additive secret sharing-based MPC [7], which represents each data value as an element in a finite algebraic structure. This structure is typically a finite (Galois) field or a $\mathbb{Z}_{2^k}$ ring. While $\mathbb{Z}_{2^k}$ rings tend to have better performance due to native integer operations, they support a limited range of arithmetic operations [56] (e.g., protocols that require a modular inverse are not supported). Existing MPC frameworks typically provide support only for a single algebraic structure. In contrast, Sequre supports both and is able to convert between different representations to achieve better performance (Additional file 1: Section 4).

There are many flavors of MPC that differ in the desired security and performance guarantees, often trading off one for the other. Sequre uses additive (arithmetic) secret sharing with a trusted dealer under an honest-but-curious security model [7], which we view as a balanced option achieving both practical efficiency and a meaningful level of security. Our framework builds upon the MPC framework used by the prior work on secure GWAS [9], which combines a variety of key MPC building blocks from the literature (e.g., for fixed-point arithmetic and comparison protocols) into a unified MPC library. Sequre supports joint computation among any number of computing parties (at least two, not including the trusted dealer). It remains secure against arbitrary collusion among parties as long as the trusted dealer and at least one other party (participating in additive secret sharing) remain honest. Since Sequre provides a general framework for MPC implementation and optimization, it allows end-users to extend and adjust the existing MPC protocols, as well as to implement novel MPC protocols that could provide different security guarantees if desired.

**MPC-specific optimizations**

Sequre performs five compile-time MPC-specific optimizations: two network load optimizations, one code generation optimization, and two low-level performance optimizations. All of them are automatically invoked via the custom `@sequre` decorator (Fig. 2; see Additional file 1: Section 4 for details).

***Network optimizations***

Sequre provides two network optimizations to reduce the communication rounds and the overall network bandwidth in the secure multiplication routine. This routine by default uses a generalized form of the Beaver multiplication triples [57], which were originally devised for secure multiplication of two elements, but later generalized for computing higher order polynomials [9]. Such computation necessitates constructing the so-called *Beaver partitions* of the secretly shared data beforehand (Additional file 1: Section 4).

A naïve implementation of this procedure calculates fresh Beaver partitions in each multiplication for each variable. However, the Beaver partitions of the past variables

that have not been modified can be reused in the future, which can significantly improve the performance of the overall pipeline because Beaver partitioning is expensive both in terms of network utilization and computational overhead [9]. Such reuses are typically implemented manually and require developers to carefully inspect the code and avoid redundant partitions when needed. While this manual optimization can significantly reduce the runtime of the protocol, it complicates the development process and makes the underlying code complex and less readable. For example, an optimized secure MPC implementation of QR factorization or a simple linear regression [9] can become 10× longer than the non-secure program implementing the same algorithm due to manual optimizations.

Sequre addresses this problem by automatically tracking the multiplication operations and finding the places where Beaver partitions can be reused through static code analysis methods described below.

*Beaver caching optimization:* As mentioned earlier, the MPC framework of Sequre requires Beaver partitions of the input variables before each multiplication. Once generated, these partitions can be cached and reused in subsequent multiplications as long as the variable remains unchanged. Furthermore, some operations, like addition and public scalar multiplication, are invariant to Beaver partitioning (i.e., when adding two numbers, it is enough to add the corresponding partitions to obtain the partitions of the sum). Hence, the partitions of the sums can be propagated and reused in subsequent multiplications, thus avoiding redundant computation and communication across different multiplications (Fig. 2). Sequre automates the partition reuse by statically analyzing arithmetic expressions that operate on secretly shared variables. Generated partitions are cached and reused by traversing the binary expression tree for each target expression and by labeling the redundant sub-expressions, identified either directly or through propagation. For a set of expressions that share the same variable, the variable is partitioned in only one expression; other expressions in the same set reuse the cached partitions. Sequre also tracks changes to variables and invalidates a cached partition whenever a change occurs.

*Polynomial optimization:* Arithmetic expressions that contain operations that rely on Beaver multiplications (multiplication, addition, and exponentiation) can often be represented in a generalized polynomial form:

$$f_m(x_1, x_2, \ldots, x_n) = \sum_{i=1}^{m} c_i \prod_{j=1}^{n} x_j^{p_{ij}}, \ c_i, x_j \in \mathbb{R}, p_{ij} \in \mathbb{N}_0,$$

where $\mathbb{N}_0$ denotes non-negative integers including zero and $\mathbb{R}$ real numbers. Certain types of these polynomials (e.g., a low-degree polynomial) can be efficiently evaluated by the generalized Beaver partitioning approach [9], where Beaver partitions of the input variables are calculated only once. However, manually formulating such polynomials from the existing expressions is a cumbersome task that often requires large-scale code changes. Even when the polynomials are identified, this procedure is hard to implement manually.

To address this problem, Sequre automatically enumerates all candidate polynomials from a block of expressions at compile-time, then identifies a sufficient set of

polynomials that can be efficiently evaluated to obtain the final results. Afterwards, it generates the secure generalized polynomial evaluation procedures for each identified polynomial (Fig. 2). By choosing the best polynomials to evaluate, Sequre minimizes the overall network overhead for evaluating a set of expressions. Note that, while our polynomial evaluation routine minimizes the rounds of communication, it introduces an offline performance overhead that can grow exponentially with the degree of the polynomial. For this reason, Sequre limits the degree of polynomials for this optimization and resorts to the Beaver caching technique to further divide the expression into smaller components if the full expansion is deemed to be infeasible.

### *Code generation and performance optimizations*

Sequre takes advantage of the specific nature of MPC algebraic operations to introduce domain-specific optimizations that can improve elementary operations such as division and square root. For example, the standard iterative technique for calculating a square root via MPC (based on Goldschmidt's algorithm) additionally outputs the inverse square root as a by-product. Thus, an efficient way of dividing by the square root of some number $b$ (e.g., normalizing a vector by its norm) is to multiply the numerator by inverse square root of $b$, as opposed to invoking both square root and division operations [9]. Sequre compiler identifies this and other similar patterns and replaces the expression with an equivalent expression that can be more efficiently evaluated under MPC (Fig. 2). In addition, Sequre tracks the diagonal matrices in the expressions using a vector representation and replaces the matrix operations involving them with efficient vector-based operations that avoid unnecessary calculations involving the off-diagonal zeros. Sequre also optimizes compute-intensive operations like matrix multiplication by using a specialized, LLVM-optimized version of the Strassen algorithm.

We also note that the modulo operator is one of the most commonly used operators in secret sharing-based MPC protocols. Unfortunately, LLVM ships with a generic implementation of the modulo operation, which often offers substandard performance. For this reason, Sequre introduces a new and efficient calculation of fixed modulus operations (as the modulus remains fixed in MPC protocols), yielding up to 40% of performance improvement over the default LLVM modulo operator (Fig. 3).

### Code and data availability

Our software suite, including documentation and tutorials, is available at https://github.com/0xTCG/sequre. Details about our datasets and experimental protocols are provided in Additional file 1: Section 5.

### Supplementary Information

---

Additional file 1. Sequre supplementary notes provides additional insight to Sequre, its usability and optimizations, and the results [61–90].

Additional file 2. Review history.

### Acknowledgements
We would like to thank Nasrin Akbari and Amirali Baniasadi at the University of Victoria for helpful suggestions and discussions.

### Peer review information

### Review history
The review history is available as Additional file 2.

### Authors' contributions
A.S., B.B., H.C., and I.N. jointly conceived the idea of Sequre. H.S. designed and implemented Sequre. A.S. and I.N. created the compiler framework that underpins Sequre. B.B. and H.C. provided a theoretical and technical baseline for secure multiparty computation protocols. All authors were the major contributors in writing the manuscript, and all authors read and approved the final manuscript.

### Funding
This work was partially supported by NSERC Discovery Grant (to I.N.), Canada Research Chair program (to I.N.), NIH R01 HG010959 (to B.B.), and NIH DP5 OD029574 (to H.C.).

### Availability of data and materials
Our software suite, including documentation and tutorials, is published under the Apache 2 license at https://github.com/0xTCG/sequre/tree/v0.0.1-alpha [58] and https://doi.org/10.5281/zenodo.7465764 [59]. The datasets supporting the conclusions of this article are included within the Additional file 1. The version of the source code used in the manuscript is v0.0.1-alpha. For lung cancer data in GWAS analysis, we used dbGaP data with accession phs000716.v1.p1 [60].

## Declarations

### Ethics approval and consent to participate
Not applicable.

### Consent for publication
Not applicable.

### Competing interests
A.S., I.N., and B.B. are shareholders of Exaloop inc., which maintains the Seq language on top of which Sequre was built.

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

## 
