## [Additional file 2. Review history. · Genome Biology]

Review History

First round of review

Reviewer 1

Were you able to assess all statistics in the manuscript, including the appropriateness of statistical tests used? Yes, and I have assessed the statistics in my report.

Were you able to directly test the methods? Yes.

Comments to author:

The practical application of MPC has been hampered by the high cost of developing efficient MPC protocols. As genomic research increasingly relies on data from multiple sources, the need to protect data privacy has become more important. However, many genomic researchers are not well-trained in secure multiparty computation (MPC), which could be used to protect data privacy. In order to use MPC efficiently, researchers need to be able to use low-level programming language and optimize code (network, caching, arithmetic operations, etc.). However, many genomic researchers do not have the necessary programming skills. This lack of training potentially exposes a gap in the responsible use and sharing of sensitive data.

Sequire offers a set of automatic compile-time optimizations that significantly improve the performance of MPC applications. These optimizations include reducing the number of operations required for each computation, and improving the efficiency of data storage and communication. As a result, MPC applications compiled with Sequire can run up to several times faster than those compiled without these optimizations.

The authors demonstrated the performance advantage of Sequire with real applications. They showed that Sequire outperforms traditional approaches in terms of efficiency. For example, Sequire-GWAS is a reimplement of the MPC-based GWAS pipeline that is composed of only 160 lines of high-level Python code. It is able to obtain a $3.7\times$ decrease in the overall runtime of GWAS while maintaining a comparable network utilization on the lung cancer dataset. Similarly, the authors demonstrated the performance advantage of Sequire Drug-Target Interaction Prediction by showing that their method outperformed existing methods in terms of speed with comparable accuracy and code length. The reviewer has the following suggestions:

It would be helpful to have a table of all the matrix operations that are supported. This would make it easier to see what is possible and to find the operation that a researcher would need. As more and more deep learning packages are supporting tensor operations, it might be interesting to include in the discussion about future plans to support efficient secure tensor operations (i.e., Tenseal for homomorphic encryption is an example).

As the genomic research community continues to grow, the demand for such MPC libraries will also increase. In order to meet this demand, it might be nice to discuss a potential plan to maintain and extend the library so that it can continue to be used by researchers.

This work is very timely in that it bridges genomic privacy research with MPC optimization. By doing so, it can help to improve the privacy of genomic data while also making it more accessible for research purposes. This is an important step in ensuring that genomic data is protected while still being available for research purposes.

Reviewer 2

Were you able to assess all statistics in the manuscript, including the appropriateness of statistical tests used? There are no statistics in the manuscript.

Were you able to directly test the methods? Yes.

Comments to author:

The paper presented the Sequire as a high-performance framework for secure Multiparty Computation (MPC) of sensitive genomic/biological data. Sequire implemented optimization strategies through compiling, which are thus transparent from the developers. The developers instead need to write python-like code without worrying about the low-level tricks for MPC algorithms. As a result, Sequire improves both the algorithmic efficiency and the usability by bioinformatics developers who are not familiar with MPC protocols. I think this is a critical step for the cryptographic approaches to be widely used by the genomics/bioinformatics community. The authors also implemented several bioinformatics applications using Sequire, showing the implementations becomes not only easier (with fewer lines of codes) but also several times faster. The authors have done a good job to describe the implementation details and compared with existing frameworks in the supplementary notes.

I have a few minor suggestions for the manuscript.

- 1) Sequire currently implemented only the (Beaver's) additive secret sharing protocol. It will be useful to discuss the application scope of the protocol from the perspective of the assumptions related to the attack model (e.g., a trusted party to perform light computation, collusions, etc), in particular in the context of bioinformatics applications.
- 2) For the application of metagenomic sequence binning, as there is no baseline MPC implementation, it is useful to present the runtime of non-secure algorithms (Ganon and Opal) to demonstrate the overhead of MPC.
- 3) The comparison with existing frameworks. Currently, the results are all in the supplementary notes (which is very long). It may be useful to include a brief table in the main paper for the comparison with some framework with similar functionalities.

Reviewer #1

The practical application of MPC has been hampered by the high cost of developing efficient MPC protocols. As genomic research increasingly relies on data from multiple sources, the need to protect data privacy has become more important. However, many genomic researchers are not well-trained in secure multiparty computation (MPC), which could be used to protect data privacy. In order to use MPC efficiently, researchers need to be able to use low-level programming language and optimize code (network, caching, arithmetic operations, etc.). However, many genomic researchers do not have the necessary programming skills. This lack of training potentially exposes a gap in the responsible use and sharing of sensitive data.

Sequire offers a set of automatic compile-time optimizations that significantly improve the performance of MPC applications. These optimizations include reducing the number of operations required for each computation, and improving the efficiency of data storage and communication. As a result, MPC applications compiled with Sequire can run up to several times faster than those compiled without these optimizations.

The authors demonstrated the performance advantage of Sequire with real applications. They showed that Sequire outperforms traditional approaches in terms of efficiency. For example, Sequire-GWAS is a reimplementations of the MPC-based GWAS pipeline that is composed of only 160 lines of high-level Python code. It is able to obtain a 3.7× decrease in the overall runtime of GWAS while maintaining a comparable network utilization on the lung cancer dataset. Similarly, the authors demonstrated the performance advantage of Secure Drug-Target Interaction Prediction by showing that their method outperformed existing methods in terms of speed with comparable accuracy and code length.

We thank the reviewer for their thoughtful comments.

The reviewer has the following suggestions:

1. It would be helpful to have a table of all the matrix operations that are supported. This would make it easier to see what is possible and to find the operation that a researcher would need.

Thank you for the suggestion; please see the response below.

2. As more and more deep learning packages are supporting tensor operations, it might be interesting to include in the discussion about future plans to support efficient secure tensor operations (i.e., Tenseal for homomorphic encryption is an example).

Thank you for suggesting this. Sequire already operates on top of the SharedTensor data structure that stores n-dimensional arrays shared across multiple computing parties (akin to OpenMined's shared tensor). We have added a new subsection 4.2.7 ("Shared tensor and supported operations") to the Additional file 1 that introduces the SharedTensor class and its functionalities. Additionally, for each SharedTensor operation, a working example is added. Finally, we have added a guide on how to use Sequire and its features, including SharedTensor and all the operations supported, to the project's wiki pages: <https://github.com/0xTCG/sequire/wiki>.

3. As the genomic research community continues to grow, the demand for such MPC libraries will also increase. In order to meet this demand, it might be nice to discuss a potential plan to maintain and extend the library so that it can continue to be used by researchers.

We have secured funding from NSERC and CRC (Canada) for this project for at least two more years. This project and its spinoffs are also funded through the NIH R01 and NIH DP5 grants Sequire development. The first author (Haris Smajlović) will work on Sequire for the duration of his graduate studies as the core maintainer; other collaborators from the Broad Institute are also working actively on developing and maintaining Sequire, as well as adapting it to their internal pipelines. Finally, the software is open-source, and that will enable wider collaboration with other interested parties.

This work is very timely in that it bridges genomic privacy research with MPC optimization. By doing so, it can help to improve the privacy of genomic data while also making it more accessible for research purposes. This is an important step in ensuring that genomic data is protected while still being available for research purposes.

We thank the reviewer for their encouraging comments.

Reviewer #2

The paper presented the Sequire as a high-performance framework for secure Multiparty Computation (MPC) of sensitive genomic/biological data. Sequire implemented optimization strategies through compiling, which are thus transparent from the developers. The developers instead need to write python-like code without worrying about the low-level tricks for MPC algorithms. As a result, Sequire improves both the algorithmic efficiency and the usability by bioinformatics developers who are not familiar with MPC protocols. I think this is a critical step for the cryptographic approaches to be widely used by the genomics/bioinformatics community. The authors also implemented several bioinformatics applications using Sequire, showing the implementations becomes not only easier (with fewer lines of codes) but also several times faster. The authors have done a good job to describe the implementation details and compared with existing frameworks in the supplementary notes.

We sincerely thank the reviewer for their valuable comments and insights.

I have a few minor suggestions for the manuscript.

1. Sequire currently implemented only the (Beaver's) additive secret sharing protocol. It will be useful to discuss the application scope of the protocol from the perspective of the assumptions related to the attack model (e.g., a trusted party to perform light computation, collusions, etc), in particular in the context of bioinformatics applications.

Sequire implements additive secret sharing schemes mostly due to their performance, as many efficient routines were developed specifically for such schemes. This performance is of crucial importance for bioinformatics applications that operate on large-scale datasets.

The trusted party only handles light preprocessing to accelerate the main protocols (e.g., Beaver's multiplication). This means that the collaborating entities must identify an independent, trustworthy actor to assume this role. However, this is already the case in all bioinformatics data-sharing agreements that are commonly facilitated by a trusted party (e.g., NIH, dbGaP, and so on). Other details have been extensively discussed in Cho et al. (2018); to that end, we have added the following clarification to the section "Sequire and other MPC frameworks" in the main article:

*It [Sequire] operates under what we view as middle-ground security constraints, providing a rigorous notion of security based on the properties of secret sharing, while introducing additional requirements to enable efficient performance, namely an auxiliary party---known as a trusted dealer---that generates correlated randomness to be used in the main protocol to greatly accelerate the computation. This model is also known as the server-aided model of MPC, **and it requires that the collaborating entities identify an independent, trustworthy actor to assume this role. However, many bioinformatics applications with large-scale datasets currently necessitate this modification to achieve practical performance. Nevertheless, disabling a trusted dealer at the expense of performance can be done, should one wish to do so. Additionally, as discussed in Cho et al., the***

existing MPC scheme can be strengthened to allow both malicious and semi-honest adversaries, which relaxes the security assumptions by allowing the parties to deviate from the protocol at the expense of having worse performance.

Finally, note that we can incorporate alternative MPC frameworks and setups that do not rely on trusted parties within Sequare, and their incorporation is planned for the future.

2. For the application of metagenomic sequence binning, as there is no baseline MPC implementation, it is useful to present the runtime of non-secure algorithms (Ganon and Opal) to demonstrate the overhead of MPC.

We thank the reviewer for this comment. To address it, we extended the "Secure metagenomic binning" section in the main article with a brief mention of the offline, non-secure runtimes of Ganon and Opal implemented in Seq language and Python, respectively. We note that, additionally, the same runtimes are added to Table 4 in Additional file 1.

3. The comparison with existing frameworks. Currently, the results are all in the supplementary notes (which is very long). It may be useful to include a brief table in the main paper for the comparison with some framework with similar functionalities.

We thank the reviewer for bringing this to our attention. We have subsampled the runtime comparisons table from Additional file 1 and added it to the main article. The runtime comparison between Sequare and the three frameworks most similar to it in terms of design and features, namely the MPyC, MP-SPDZ, and Sharemind, were added to the table at the end of the section "Sequare and other MPC frameworks" within the main article.